# Microfluidic Diffusion Sizing Applied to the Study of Natural Products and Extracts That Modulate the SARS-CoV-2 Spike RBD/ACE2 Interaction

**DOI:** 10.3390/molecules28248072

**Published:** 2023-12-13

**Authors:** Jason Fauquet, Julie Carette, Pierre Duez, Jiuliang Zhang, Amandine Nachtergael

**Affiliations:** 1Unit of Therapeutic Chemistry and Pharmacognosy, University of Mons (UMONS), 7000 Mons, Belgium; jason.fauquet@umons.ac.be (J.F.); pierre.duez@umons.ac.be (P.D.); amandine.nachtergael@umons.ac.be (A.N.); 2College of Food Science and Technology, Huazhong Agricultural University, Wuhan 430070, China; zjl_ljz@mail.hzau.edu.cn

**Keywords:** hydrodynamic radius, dissociation constant, protein-protein interaction

## Abstract

The interaction between SARS-CoV-2 spike RBD and ACE2 proteins is a crucial step for host cell infection by the virus. Without it, the entire virion entrance mechanism is compromised. The aim of this study was to evaluate the capacity of various natural product classes, including flavonoids, anthraquinones, saponins, ivermectin, chloroquine, and erythromycin, to modulate this interaction. To accomplish this, we applied a recently developed a microfluidic diffusional sizing (MDS) technique that allows us to probe protein-protein interactions via measurements of the hydrodynamic radius (R_h_) and dissociation constant (K_D_); the evolution of R_h_ is monitored in the presence of increasing concentrations of the partner protein (ACE2); and the K_D_ is determined through a binding curve experimental design. In a second time, with the protein partners present in equimolar amounts, the R_h_ of the protein complex was measured in the presence of different natural products. Five of the nine natural products/extracts tested were found to modulate the formation of the protein complex. A methanol extract of *Chenopodium quinoa* Willd bitter seed husks (50 µg/mL; bisdesmoside saponins) and the flavonoid naringenin (1 µM) were particularly effective. This rapid selection of effective modulators will allow us to better understand agents that may prevent SARS-CoV-2 infection.

## 1. Introduction

It is difficult to ignore the global health crisis caused by severe acute respiratory syndrome coronavirus 2 (SARS-CoV-2). First appearing in Wuhan, China, in December 2019, this single-stranded positive sense RNA virus led to an estimated 771 million infections and 6.9 million deaths [1]. Currently, although it has attenuated, the 2019 coronavirus disease (COVID-19) pandemic continues worldwide, with many variants arising. Thus, the need to minimize the spread of the virus and to find rapid and effective prevention and treatment means has pushed scientists to expand their knowledge about the disease considerably.

Even though the methods of transmission are multiple, the major means of dispersion are still through the respiratory tract [2,3]. The key factor for viral invasion in humans relies on an interaction between the receptor binding domain (RBD) of the spike protein (S protein) and human angiotensin-converting enzyme 2 (ACE2); this interaction initiates membrane fusion, allowing virus entry, and it is involved in syncytia formation, potentially implicated in viral dissemination and the immune response [4,5,6].

The well-known “crown-like” appearance from which the family name was inspired seems to belong to the presence of two conformations of “spike” proteins on the surface of mature virion. The “prefusion” conformation is first produced as a trimerized precursor composed of three protomers [7], each protomer being composed of two subunits: the S1 carrying the RBD domain mainly involved in the interaction with the ACE2 receptor; and the S2 anchoring the proteins into the membrane. The RBD domain standing at the apex of each protomer adopts two distinct “prefusion” conformations, called “up” if the receptor is accessible and “down” if it is not [8]. This position varies from one prefusion S protein to another, adopting, for example, one RBD “up” and two “down”. Inducing the fusion process requires protein S cleavage, leading to a new conformation, “postfusion”, consisting of a loss of S1 subunits and refolding of S2 subunits. This new conformation also appears in the absence of interaction with ACE2, which explains the presence of these two conformations at the surface of the mature virion [7,9,10,11].

ACE2, the human-functional receptor for SARS-CoV-2, has several roles, ranging from enzymatic activities with various substrates to amino acid transporter partners, but its main function is to degrade angiotensin II into angiotensin 1–7 [8,12], playing a key role in maintaining salt and water homeostasis. ACE2 is distributed in the lungs, cardiovascular system, gut, kidneys, central nervous system, and adipose tissue. However, the highest ACE2 concentrations are found in the lungs, in which 83% of the cells expressing ACE2 are type 2 alveolar cells of the pulmonary epithelium [13,14]. The ACE2 protein is composed of 805 amino acids and has two functional parts, an N-terminal domain containing a peptidase M2 function and a C-terminal collectrin domain; the human ACE2 peptidase domain interacts with the RBD of the viral spike protein. This interaction leads to a conformational change in the S-protein, allowing for fusion with the host cell membrane. This binding ACE2-RBD has been shown to be a relevant target to prevent infection [15]. In the present paper, we applied a promising method based on microfluidic diffusional sizing (MDS) to highlight several compounds able to modulate and disrupt this interaction [16].

The MDS method, initially introduced in 2016 [17], has garnered over 111 citations, indicating its swift emergence as a versatile technique, notably for the identification and quantification of protein-protein interactions in various mixtures. The versatility of MDS positions it for a broad spectrum of applications, including the analysis of lipids, proteins, nucleic acids, and proteins, the sizing of single molecules, and the monitoring of biomolecule assembly, for example, antigen to antibody or protein to chaperone.

Natural products provide original and diversified insights to develop a therapeutic arsenal against viral infections, notably COVID-19; indeed, they offer a remarkable heterogeneity of chemical structures capable of targeting different stages of the SARS-CoV-2 life cycle [18,19,20]. Several classes of natural products have notably been considered to modulate the spike RBD/ACE2 interaction. Among them, some deserve further research, as they could do the following:

✓Show interesting binding energy to one or both protein partners (flavonoids [18,21,22,23,24,25,26,27], terpenoids [28,29,30,31,32,33,34], macrocycles [5,35,36,37,38], alkaloids [28], and anthraquinones [32,39,40]).✓Disrupt the spike RBD/ACE2 complex, bind the spike protein, or inhibit ACE2 activity (polyphenols such as flavonoids [23,24,41], stilbenes [42,43] or tannins [44,45]; terpenoids [46,47], and anthraquinones [48]).✓Prevent viral entry into cells expressing ACE2 (flavonoids [49], cannabinoids [50], terpenoids [51,52], quinones [53], macrocycles [54], alkaloids [55,56], and anthraquinones [57]).

These previous data served as the basis for the selection of the varied structures studied in the present work for their impact on spike RBD/ACE2 binding, i.e., flavonoids, anthraquinones, saponins, ivermectin, chloroquine, and erythromycin.

## 2. Results

### 2.1. SARS-CoV-2 Spike RBD/ACE2 Binding: Determination of K_D_

The dissociation constant (K_D_) of the SARS-CoV-2 spike RBD/ACE2 complex was measured over a range of ACE2 concentrations (180 pM to 750 nM) for a spike RBD concentration set at 20 nM; these concentrations were selected to yield two distinct plateaus. The R_h values_ measured for free spike RBD and the spike RBD/ACE2 complex at saturation are 2.97 nm (95% confidence interval, 2.90–3.05 nm) and 4.35 nm (95% confidence interval, 4.21–4.50 nm), respectively. Fitting of the obtained sigmoid curve (Figure 1) yielded a K_D_ of 24.2 nM (95% confidence interval, 10.6–48.9 nM). The obtained K_D_ is within the range of the previously published K_D_ (Table 1).

### 2.2. Validation of MDS Measurements

Given the relative recentness of the MDS method, a series of validation parameters were assessed, including selectivity, within- and between-chip reproducibility, accuracy, and precision.

The microdiffusion chips were proposed as single-use chips, and their eventual reuse was tested but found to be unreliable (Appendix A) as repeated measurements do not agree, preventing the determination of a K_D_ over a single chip. The criteria of selectivity, accuracy, precision, and quality of adjustment were then determined by running a single injection per chip. A summary of the validation data is presented in Table 2, while full data are reported in the Appendix A.

### 2.3. Modulation of SARS-CoV-2 Spike RBD/ACE2 Binding by Natural Compounds

First, a day-to-day variation was observed for the R_h_ of the SARS-CoV-2 spike RBD/ACE2 complex itself; this could be related to a difference in chip temperature that is unavoidable, given that the system is not temperature controlled and depends on the laboratory environmental conditions. Consequently, daily controls were measured for the SARS-CoV-2 spike RBD/ACE2 complex in the absence of natural products; all data obtained in the presence of natural products were then compared to their daily controls. To minimize the effect of an eventual temperature change during daily experiments, the controls and samples were randomly analyzed within the same day.

Table 3 presents the modulation of the R_h-complex_ when proteins are incubated in the presence of natural products. Quercetin reduces the R_h-complex_ of SARS-CoV-2 Spike RBD/ACE2 by 11% at a concentration of 150 μM, while naringenin has an impact on the R_h-complex_ at a concentration of only 1 μM with a significant positive R_h-complex_ variation of approximately 10%.

The literature data indicate promising in vitro activities for ivermectin. In our MDS model, ivermectin induces a significant increase in the R_h-complex_ by approximately 12% at a concentration of 1 nM. This effect seems to be concentration-dependent over the tested range. The dry extract of Rhei radix is active at a concentration of 100 μg/mL with an Rh_-complex_ decrease of approximately 14%. The saponin-rich extract of *Chenopodium quinoa* husks has shown the most interesting activity; a modulatory effect was already statistically significant at a concentration of 50 μg/mL with an increase in the R_h-complex_ by approximately 40%. The observed effect appears dose-dependent, with an R_h-complex_ increase of approximately 150% at a concentration of 200 μg/mL. Finally, natural products that were noneffective in our experimental design are summarized in Table 4.

## 3. Discussion

As shown in Figure 2 and Figure 3, the binding between SARS-CoV-2 spike RBD and ACE2 could be modulated in two opposite ways by the tested natural compounds:

An increase in the hydrodynamic radius (higher R_h-complex_) that could correspond either to a distension of the bound protein complex or to a clustering of natural compounds on the proteins; these should shift the affinity curve to the left and decrease K_D_.A decrease in the hydrodynamic radius (lower R_h-complex_), possibly indicating a collapse or folding of the complex or a partial separation of the two protein partners; this should shift the affinity curve to the right and increase K_D_.

Although it is difficult to define the impact of these modulations on virus entry, the MDS method has the advantage of quickly sorting natural compounds and extracts to pinpoint interfering phytochemicals that can then be evaluated in more detail.

### 3.1. Quercetin

The online *Traditional Chinese Medicine Systems Pharmacology (TCMSP) Database and Analysis Platform* (available at https://old.tcmsp-e.com/tcmsp.php; [71]; accessed on 30 October 2023) predicts quercetin as a potential key drug. Additionally, a molecular docking analysis by the software AutoDock 4.2 predicts binding energies of −7.92 kcal/mol to ACE2 and −8.41 kcal/mol to apike RBD [24]; such a value < −6.2 kcal/mol indicates a possible modulation of protein-protein binding [24,35,38,42,44,72].

The obtained results partly agree with the literature data; indeed, Pan et al. (2020), applying surface plasmon resonance, reported a strong modulation of the interaction, but at a much lower concentration of quercetin (12 µM). Although the concentration found effective here (>30 µM) may be toxic under physiological conditions [73], our results point to an interest in the research of molecules with related structures that would be active at low concentrations.

### 3.2. Naringin and Naringenin

The flavanone naringenin, selected for its binding energy to ACE2 of −6.05 kcal/mol [27] and its ability to inhibit, in a dose-dependent manner, the entry of SARS-CoV-2 in Vero E6 cells [49], was found to be highly effective as a modulator of SARS-CoV-2 spike RBD/ACE2 binding (active at 1 µM). Concentrations of 15.7 µM (C_Max_) could be achieved within 3.2 h after oral administration of 150 mg of naringenin (in 536 mg of *Citrus sinensis* extract); no adverse events or changes in blood markers were detected following the oral administration of a naringenin dose up to 900 mg [74].

In contrast, its heteroside, naringin, had no significant effect on the R_h-complex_ despite its reported binding energy to ACE2 (−6.85 kcal/mol) [22]. This can probably be explained by steric hindrance of its O-rhamnoglucoside at position 7. Indeed, the predicted binding sites are significantly different for naringin (TYR-515, GLU-402, GLU-398, and ASN-394) and naringenin (PRO-146, LEU-143, and LYS-131) [22,27]; interestingly, our data indicate that the naringenin interaction site would be important for SARS-CoV-2 spike RBD/ACE2 binding.

### 3.3. Ivermectin

The literature data indicate promising in vitro activities for ivermectin; applied 2 h postinfection, ivermectin decreases the amount of viruses in infected Vero-hSLAM cells [54], while in silico modeling indicates ACE2 and spike RBD binding energies of −18 kcal/mol and −10.87 kcal/mol, respectively [37,38]. Despite this, a series of clinical studies did not yield conclusive data, except for the Biber et al. (2022) study, which showed a reduction in viral transmission in the frame of a clinical trial [75]. Our results are consistent with the in vitro data from the literature and place ivermectin on the side of interesting molecules that seem worthy of further investigation. Moreover, published data indicate that 23 to 27 nM (C_Max_) could be achieved in plasma within 4.5 h after administration of a 6 mg tablet [76].

### 3.4. Rhei Radix

The dry extract of Rhei radix (*Rheum palmatum* L., *Rheum officinale* Baillon) was selected for its content in anthraquinones, notably rhein and emodin. Rhein has a predicted ACE2 binding energy of −8.73 kcal/mol [39], and emodin has binding energies to ACE2 and spike RBD of −7.26 kcal/mol and −8.8 kcal/mol, respectively [32,40]. Moreover, emodin has been reported to inhibit the entry of SARS-CoV-2 in Vero E6 cells [57]. Our first effective concentration corresponded to 3.6 µg/mL rhein, the only anthraquinone component of rhubarb that is absorbed into the blood in the human body [77]. In human volunteers, a single dose of rhubarb extract (50 mg/kg, corresponding to approximately 54 mg hydroxyanthracene derivatives, body weight 60 kg) [78] yielded a C_Max_ of 2.8 µg/mL within approximately 59 min. This indicates that the active concentration of hydroxyanthracene glycosides cannot be yielded in plasma and would rather favor a nasal administration of glycosides for a local preventive effect.

### 3.5. Bitter Chenopodium Quinoa Husks

Antiviral effects of some saponins have already been shown on different respiratory viruses, including the influenza virus, the human respiratory syncytial virus, and some coronaviruses [79]. For the latter, notably,

In silico docking studies predict the following:Stigmastane-type steroidal saponins (vernonioside A2, vernonioside A4 and vernonioside D2) exhibit inhibitory potential against SARS-CoV-2 cysteine proteases [80].Saponins are potential inhibitors of the SARS-CoV-2 main protease (M^pro^) with favorable ADMET profiles, with thirteen [81] and three (arjunic acid, thesapogenol B, euscaphic acid) [82] high potency compounds identified.Glycyrrhizin has the potential to bind the host cell ACE2 receptor [83], and saikosaponin, glycyrrhizin, and ilexgenin A can bind both ACE2 and the SARS-CoV-2 main protease [84].A series of saikosaponins favorably bind to the RDB region of the SARS-CoV-2 spike protein, with saikosaponin B4 as the best probable inhibitor [31], and saikosaponins bind to the NSP15 endoribonuclease and to the prefusion spike glycoprotein SARS-CoV-2, saikosaponins U and V, showing the highest affinity toward both proteins [33].In vitro studies indicate the following:Oleanane saikosaponin B2, at 6 µM, significantly inhibits viral attachment and penetration, impeding HCoV 229E infection in pre-, co-, and postinfection models [85].Glycyrrhizin at 600 µg/mL (EC50) inhibits the replication of 2 SARS-CoV clinical isolates in Vero cells [86] but was deemed inactive (EC50 > 400 µg/mL) when tested against 10 clinical strains of SARS-CoV in the fRhK4 cell line [87].Derivatives of glycyrrhizic acid are 10 to 70 times more active than glycyrrhizin itself in inhibiting the replication of a SARS-CoV clinical isolate in Vero cells; however, some compounds lose advantages in terms of viral selectivity [88].Aescin (6 µM) and four glycyrrhizin and aescin derivatives (<100 μM) showed activities toward SARS-CoV (H.K. strain) in Vero cells [89].

The dry methanolic extract of *C. quinoa* husk has a very high content of saponins, mostly bisdemosides (Figure 3); their structures are quite similar to those of the bisdesmosides saikosaponins U and V, which, in silico, present a high affinity toward the prefusion spike glycoprotein SARS-CoV-2 [33].

Regarding the first effective concentration (corresponding to 15 µg saponins/mL, expressed as hederacoside C), the poor intestinal absorption of saponins should be noted, which is mainly due to their unfavorable physicochemical traits (molecular mass, high hydrogen-bonding capacity, and high molecular flexibility) that underlie poor membrane permeability [90]. As the active concentration of bisdesmoside saponins will probably not be achievable in plasma, nasal administration should be favored for a local preventive effect.

### 3.6. Other Natural Products Tested

Chloroquine was selected because its in silico binding energy with the ACE2 protein was predicted to be −6.45 kcal/mol [28] and because of its inhibition of SARS-CoV-2 replication in Vero E6 cells [56]. Erythromycin was selected for its very high in silico affinity toward the SARS-CoV-2 spike RBD-His protein (−9 kcal/mol) [5].

Although several flavonoids have the ability to modulate protein-protein interactions [22,24,49,72], the dry extract of *Ginkgo biloba* L. leaves did not show any effect on the R_h_ of the studied protein complex even at high concentrations (200 μg/mL, corresponding to ~50 µg/mL flavonoids and 10.6 µg/mL terpene lactones, i.e., bilobalide and ginkgolides).

## 4. Materials and Methods

### 4.1. Tested Natural Products

A series of natural products were tested (Figure 4) for their capacity to modulate the spike RBD/ACE2 interaction, including 3 flavonoids (quercetin hydrate (≥95%, Sigma-Aldrich), Merck, Hoeilaart, Belgium); naringenin (≥98%, Carl Roth, Karlsruhe, Germany); naringenin rutinoside (naringin) (European Pharmacopoeia, EDQM, Strasbourg, France, CRS batch 1.0 id 004UT0)), ivermectin (88.2% B_1a_ and 2.1% B_1b_, Sigma-Aldrich), chloroquine diphosphate (≥98%, Sigma-Aldrich), erythromycin (95.9%, Febelcare, Sint-Niklaas, Belgium), a dry ethanolic extract of Rhei radix (*Rheum palmatum* L., *Rheum officinale* Baillon; 5.69% hydroxyanthracene glycosides, expressed as rhein-8-glucoside, Conforma, Destelbergen, Belgium, batch 13D30/V54409), a reference dry extract of *Ginkgo biloba* L. leaves (European Pharmacopoeia, CRS Ginkgo dry extract for peak identification Y00010121; content in bilobalide, ginkgolide A, ginkgolide B, and ginkgolide C, 2.4%, 1.5%, 0.7%, and 0.7%, respectively; content in flavonoids, 22 to 27%), and a bitter seed husks. For the latter extract, 150 mg of *Chenopodium quinoa* Willd husks powder (0.5 mm) and 5 glass marbles (1 mm diameter) were mixed with 3.0 g of methanol 99% in a Mixer mill 400 (Retsch, Haan, Germany) (30 Hz, 10 min) and centrifuged (4000× *g*, 40 min, RT). The supernatant was evaporated to dryness under low pressure using a rotary evaporator (50 °C, 80 rpm) [91].

Each product or extract was dissolved in anhydrous dimethyl sulfoxide (DMSO-Thermo Fisher Scientific, Dilbeek, Belgium) and diluted as needed with phosphate-buffered saline at pH 7.4 without calcium, magnesium, and phenol red (PBS; 10010023-Thermo Fisher Scientific).

To the best of our knowledge, no drug targeting the spike RBD/ACE2 interaction is currently used in the clinic [92]. While the widely cited chloroquine, hydroxychloroquine, and ivermectin appeared somewhat deceiving in clinical trials, the potential of spike RBD/ACE2 modulators is such [93] that it remains worthwhile to define possibly useful compounds. Ivermectin is included in our tested natural products and can be considered a “reference drug”.

**Figure 4 molecules-28-08072-f004:**
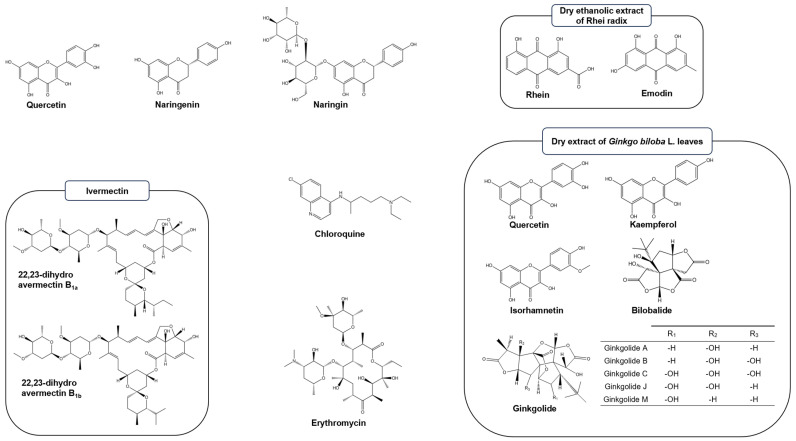
Chemical structures of the tested natural products. The figure depicts the major compounds present in tested commercial extracts [91,94,95,96,97]; their actual levels were inferred from the analysis certificates issued by the selling companies. For the extracts of Rhei radix and *Ginkgo biloba* L., chemical structures of major compounds are presented to exemplify their complex compositions [96,98]. For Chenopodium quinoa bitter seed husks, compounds were identified and quantified in μg HCe/g (μg hederacoside C equivalents per g of dry weight) by LC-MS-MS according to [91] (mean ± standard deviation; n = 3; limit of detection, 0.83 µg/g; limit of quantification, 2.53 µg/g). PA: phytolaccagenic acid; Hed: hederagenin; SA: serjenic acid; AG489 and AG487 refer to aglycones with a specific m/z; Glc: glucose; Ara: arabinose; Gal: galactose, Xyl: xylose; Hex: hexose; Pent: pentose.

### 4.2. Proteins and Fluorescence Labeling

All manipulations involving labeled proteins were protected from actinic light.

The original strain-His-tagged RBD protein and the soluble ACE2-Fc fusion protein were obtained from InvivoGen (Toulouse, France).

The RBD protein was reconstituted with PBS at a concentration of 333.3 µg/mL, added to Alexa Fluor 647 NHS ester (Thermo Fisher Scientific) at a dye-to-protein molar ratio of 3:1 and incubated at 4 °C overnight. The free dye was removed using the Antibody Conjugate Purification Kit (Thermo Fisher) following the manufacturer’s instructions. The labeled purified protein was stored at −80 °C in PBS pH 7.4 containing 10% (*v*/*v*) glycerol for a maximum of 3 months.

### 4.3. SARS-CoV-2 Spike RBD/ACE2 Affinity Measurement by Microfluidic Diffusional Sizing

The binding between ACE2 and spike RBD proteins (affinity) was measured from the value of the equilibrium dissociation constants (K_D_), obtained through a Fluidity ONE-W platform based on microfluidic diffusional sizing (MDS) and recently developed by Fluidic Analytics (Cambridge, UK) [17]. As shown in Figure 5, this method relies on microchips traveled by 2 laminar flows, parallel to each other and in direct contact, each flow leading to a fluorescence detector; the labeled species and the unlabeled (potential) partner are injected in the first flow, while the second flow is composed of a blank buffer. Due to a simple diffusion mechanism, species migrate from one flow to the second, depending on their hydrodynamic radii. The ratio of fluorescence intensities measured in each flow allows us to determine the diffusional coefficient *D* (m^2^/s) through a proprietary equation [99]; the Stokes–Einstein equation establishes the relationship between *D* and the hydrodynamic radius *R_h_*:$$D=\frac{kT}{f}=\frac{kT}{6\pi \eta R_{h}},$$
where *D* is the diffusional coefficient (m^2^/s); *k* is the Boltzman constant; *T* is the temperature (K); η is the viscosity (Pa × s); *f* is the fractional coefficient for a solid sphere in a viscous medium; and *R_h_* is the hydrodynamic radius.

Measuring *R_h_* at increasing concentrations of the partner species yields a sigmoid curve that corresponds to the saturation of the labeled species by its partner, enabling the determination of the dissociation constant (K_D_), either graphically or through a nonlinear least-squares fitting method [16].

Alexa Fluor 647-labeled spike RBD (20 nM) was mixed with unlabeled ACE-2 at increasing concentrations (180 pM to 750 nM) in a solution of DMSO 1% *v*/*v* in PBS and incubated at room temperature for 60 min. To assess the formation of complexes via MDS, 5 μL of sample was transferred to a microfluidic chip (100009; Fluidic Analytics, Cambridge, UK), and the hydrodynamic radius of bound proteins (=R_h-complex_) was determined at λ_excitation_ = 630 nm; λ_emission_ = 694 nm; and λ_dichroic_ = 660 nm with a size range of 0.7 to 5 nm (1 to 200 kDa).

### 4.4. Modulation of SARS-CoV-2 Spike RBD/ACE2 Affinity by Natural Compounds

To screen potential modulators of SARS-CoV-2 spike RBD-ACE2 receptor binding, the impact of a series of natural compounds on the R_h-complex_ was measured at defined partner concentrations via MDS. Indeed, focusing on the R_h-complex_ is better suited to screening than K_D_ determination, which would require multiple experiments at varying concentrations of both the modulator and the unlabeled partner.

To determine eventual increases or decreases in R_h_ in the presence of the tested modulators, aliquots of the 2 proteins were mixed with each potential modulator at increasing modulator concentrations to yield protein concentrations of 20 nM (equimolar concentrations near the K_D_ conditions). The mix of proteins, with and without modulators, was dissolved in a solution of DMSO 1% *v*/*v* in PBS to ensure the solubility of tested natural compounds and extracts; the solutions were incubated at room temperature for 60 min and analyzed via MDS. The spike RBD/ACE2 K_D_ determined in DMSO 1% *v*/*v* in PBS (19.8 nM) fairly agrees with the K_D_ determined in PBS (24.2 nM; 95% confidence interval, 10.6 to 48.9 nM; n = 4), indicating no detrimental effect of DMSO on measured values.

### 4.5. Statistical Analysis of Results

R_h_ values obtained for each of the tested natural product concentrations were compared to their daily negative control (R_h_ for the 20 nM equimolar mix of spike RBD and ACE2) by means of one-way ANOVA with post hoc Student’s t test (Bonferroni correction) using GraphPad Prism 5 (GraphPad, San Diego, CA, USA) software, and *p* values < 0.05 were considered significant.

## 5. Conclusions

The present study investigated the potential of selected natural products to modulate the SARS-CoV-2 spike RBD/ACE2 interaction. Using microfluidic diffusional sizing to measure the hydrodynamic radius of the bound protein complex, the effects of selected natural compounds and extracts were measured, either positive (corresponding to a distension of the complex or to a clustering of natural compounds on proteins) or negative (a collapse or folding of the complex or a partial separation of the two proteins). These data allowed us to highlight several possibly interesting compounds/extracts.

Further investigations of these compounds should include a determination of their effects on K_D_ and affinity curves, an in silico study of the effective binding sites on the protein complex, and an evaluation of their ability to prevent the cellular entry of SARS-CoV-2 in a relevant cell model.

Notably, a quinoa extract would be interesting to further study for its possible application as an infection-preventing nasal spray; although saponins are known for their sternutatory activity, the low active concentrations needed and the higher polarity and lower amphiphilicity of bisdesmosides may make them low-irritating agents to the nasal mucosa.

Compared to surface plasmon resonance and biolayer interferometry, in which a protein is immobilized on a surface with an often unknown orientation [58,59,60,61,62,63,65,70], the MDS model has the distinct advantage of working with both protein partners in a free state, with full conformational freedom, which may explain some differences from published K_D_ and modulatory effects. The development of a memtein model would be an attractive method to study, via MDS, the RBD/ACE2 interaction on an ACE2 protein in its native state, i.e., embedded in its cell membrane [102].

## Figures and Tables

**Figure 1 molecules-28-08072-f001:**
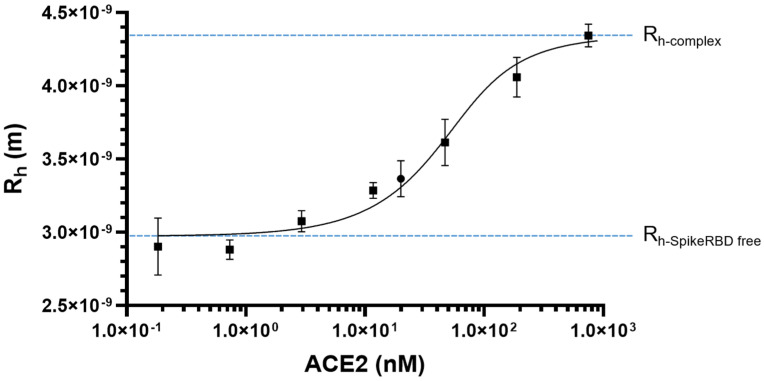
Microfluidic diffusional sizing determination of K_D_ for the spike RBD (20 nM)/ACE2 complex (DMSO, 1% *v*/*v*; room t°). R_h_ as a function of ACE2 concentration. ACE2, 180 pM to 750 nM; spike RBD, 20 nM; mean ± standard deviation (*n* = 4).

**Figure 2 molecules-28-08072-f002:**
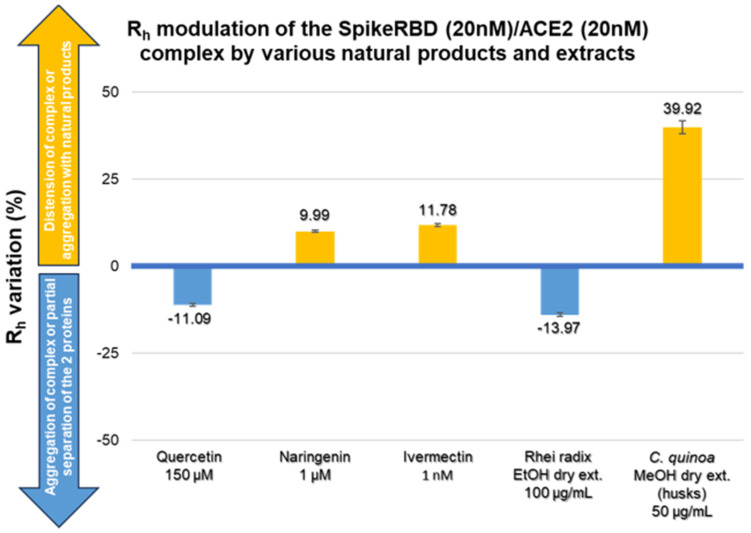
R_h_ modulation of the spike RBD (20 nM)/ACE2 (20 nM) complex by various natural products and extracts. R_h_ variation (%) of the spike RBD/ACE2 complex as a function of the tested natural products and extracts. Microfluidic diffusional size. Mean ± standard deviation. *n* = 3.

**Figure 3 molecules-28-08072-f003:**
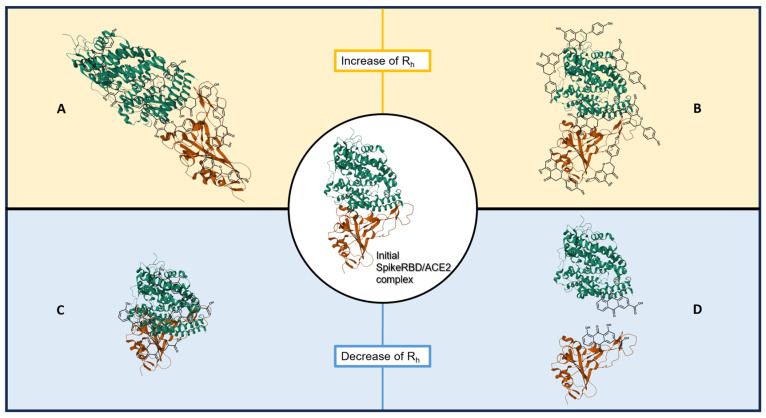
Schematic view of possible 3D conformational changes that the spike RBD/ACE2 complex could adopt in the presence of natural compounds, with their influence on R_h_. The initial conformation of the spike RBD/ACE2 complex is represented in the center (PDB 6M0J); SARS-CoV-2 spike RBD is depicted in orange and ACE2 in green. (**A**) An increase in R_h_ could be due to distension of the complex; (**B**) An increase in R_h_ could be due to clustering of natural compounds on proteins; (**C**) A decrease in R_h_ could be due to collapse or folding of the complex; (**D**) A decrease in R_h_ could be due to (partial) separation of the 2 proteins.

**Figure 5 molecules-28-08072-f005:**
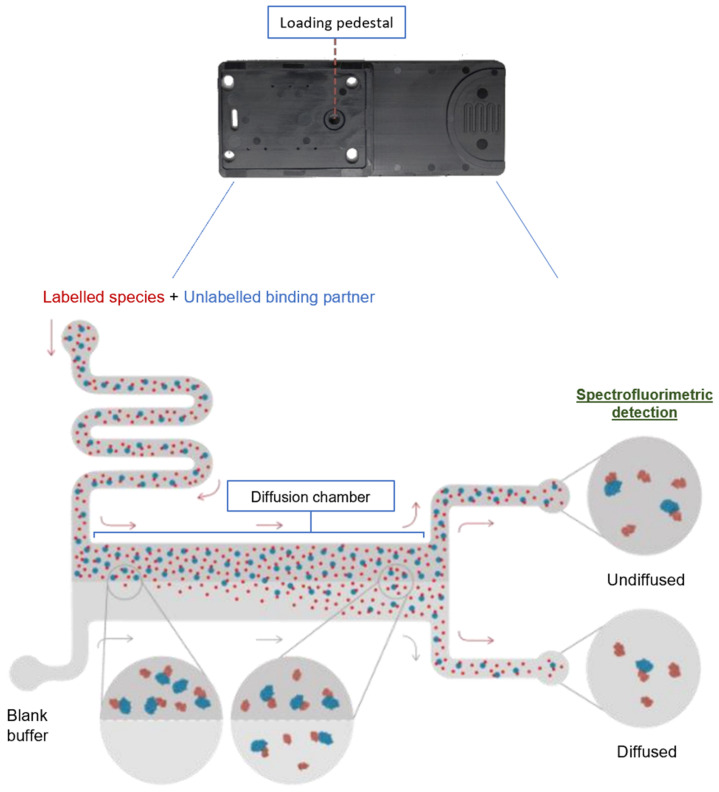
Diagram showing the interior of a microfluidic chip and the travel of the sample after injection on the loading pedestal (adapted from [100,101]).

**Table 1 molecules-28-08072-t001:** Interaction between ACE2 and SARS-CoV-2 RBD or SARS-CoV-2 spike. Published KD as measured by surface plasmon resonance (SPR) or biolayer interferometry (BLI).

Ligand-Receptor	K_D_ Determination Method	K_D_ (nM)	Reference
SARS-CoV-2 RBD/ACE2	SPR	4.7	[58]
	SPR	63.0	[59]
	SPR	24.1	[24]
	SPR	17.0	[60]
	SPR	5.8	[61]
	SPR	44.2	[62]
	BLI	75.1	[63]
	BLI	161	[64]
	BLI	172	[44]
SARS-CoV-2 Spike/ACE2	SPR	14.7	[65]
	SPR	29.1	[66]
	SPR	76	[67]
	BLI	12.8	[68]
	BLI	1.2	[69]
	BLI	133.0	[70]

**Table 2 molecules-28-08072-t002:** Summary of the validation data.

Criterion	What Is Assessed?	Result	Full Data (Appendix A)
Selectivity	Eventual fluorescence interferences	No	Appendix A
Between-chips reproducibility	1 chip for 1 R_h_ measurement (n = 6)	CV = 3.99%	Appendix A
Within-chip reproducibility	1 chip for R_h_ measurement X times at the same concentration (n = 8)	CV = 31.4%	Appendix A
Determination of K_D_ over a single chip	1 chip for 1 K_D_ determination (7 points) (n = 3)	Erroneous values	Appendix A
Quality of adjustment	Coefficient of determination (R^2^)	0.854–0.963	Appendix A
Accuracy of R_h_ for the Spike RBD_labelled_	Comparison of experimental R_h_ with a value predicted from a range of protein standards with globular conformation	105.4 ± 5.7% (n = 13)	Point 3
Precision	R_h_—Intraday precision (n = 4)	CV = 2.90%	Appendix A
	R_h_—Total precision (n = 6)	CV = 6.27%	Appendix A
	K_D_—Total precision (n = 5)	CV = 20.8%	Appendix A

**Table 3 molecules-28-08072-t003:** Natural products (NPs) effectively modulate the R_h_ of the SARS-CoV-2 spike RBD/ACE2 protein complex.

Natural Product (NP)	Quercetin		Naringenin		Ivermectin		Rhei Radix EtOH Dry Extract		*Chenopodium quinoa* Willd. MeOH Dry Extract (Husks)	
	Range Tested	1st Effective Concentration	Range Tested	1st Effective Concentration	Range Tested	1st Effective Concentration	Range Tested	1st Effective Concentration	Range Tested	1st Effective Concentration
	0.1–150 µM	150 µM	0.1–50 µM	1 µM	1–100 nM	1 nM	1–100 µg/mL	100 µg/mL	1–200 µg/mL	50 µg/mL
R_h_ of daily control (nm) (mean ± SD)	3.28 ± 0.12		2.91 ± 0.05		3.19 ± 0.07		3.03 ± 0.08		2.14 ± 0.02	
R_h_ in the presence of NP (nm) (mean ± SD)	3.35 ± 0.07 to 2.92 ± 0.03	2.92 ± 0.03	3.09 ± 0.05 to 3.21 ± 0.07	3.21 ± 0.07	3.57 ± 0.07 to 4.09 ± 0.08	3.57 ± 0.07	3.03 ± 0.07 to 2.61 ± 0.05	2.61 ± 0.05	2.34 ± 0.22 to 5.33 ± 0.31	3.0 ± 0.1
Direction of R_h_ variation	Decrease		Increase		Increase		Decrease		Increase	
R_h_ variation (%) (mean ± SD)	2.12 ± 0.09 to −11.09 ± 0.43	−11.09 ± 0.43	6.08 ± 0.15 to 19.01 ± 0.85	9.99 ± 0.29	11.78 ± 0.40 to 27.97 ± 0.82	11.78 ± 0.40	−0.15 ± 0.01 to −13.97 ± 0.46	−13.97 ± 0.46	9.19 ± 0.87 to 148.98 ± 8.76	39.92 ± 1.91

Daily control was the mix spike RBD (20 nM)/ACE2 (20 nM) analyzed on the same day as the concerned NP. The first effective concentration is the concentration yielding a significant difference from the R_h_ of the daily control (n = 3; one-way ANOVA with post hoc Student’s *t* test, Bonferroni correction; *p* < 0.05).

**Table 4 molecules-28-08072-t004:** Natural products (NP) that do not modulate the R_h_ of the SARS-CoV-2 spike RBD/ACE2 protein complex.

Natural Product (NP)	Naringin	Chloroquine	Erythromycin	*Gingko biloba* L. Dry Extract (Leaves)
Range tested	0.1–50 µM	1–1000 µM	0.1–50 µM	1–200 µg/mL
R_h_ of daily control (nm) (mean ± SD)	3.31 ± 0.06	2.87 ± 0.04	3.14 ± 0.05	2.39 ± 0.11
R_h_ of NP (nm) (mean ± SD)	3.26 ± 3.38–3.46 ± 0.12	2.76 ± 0.13–2.94 ± 0.03	3.06 ± 0.07–3.11 ± 0.08	2.35 ± 0.03–2.34 ± 0.16
R_h_ variation (%) (mean ± SD)	−1.64 ± 0.03–4.5 ± 0.2	−3.92 ± 0.20–2.11 ± 0.03	−2.29 ± 0.06–−0.73 ± 0.02	−1.75 ± 0.08–−2.10 ± 0.17
*p* Value	>0.9999–0.1256	0.4134–>0.9999	0.8427–>0.9999	>0.9999–>0.9999

Daily control was the mix Spike RBD (20 nM)/ACE2 (20 nM) analyzed on the same day as the concerned NP. None of the tested concentrations yielded a significant difference from the R_h_ of the daily control (n = 3; one-way ANOVA with post hoc Student’s *t* test, Bonferroni correction; *p* > 0.05).

## Data Availability

Data are contained within the article and Appendix A.

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
