# Peer review of "Microfluidic Diffusion Sizing Applied to the Study of Natural Products and Extracts That Modulate the SARS-CoV-2 Spike RBD/ACE2 Interaction"

_molecules, 2023, doi:10.3390/molecules28248072_

Round 1
Reviewer 1 Report
Comments and Suggestions for Authors
This manuscript reports the natural products and extracts that modulate the SARS-CoV-2 SpikeRBD/ACE2 interaction via the protocol of Microfluidic Diffusional Sizing.
The subject of study is interesting. However, I have several concerns:
1
The Microfluidic Diffusional Sizing method is used for testing.
Please review the Microfluidic Diffusional Sizing method, and provide sufficient background for this method in the Introduction section.
2
Also, if possible, please illustrate in the Materials and Method the fundamentals of how Microfluidic Diffusional Sizing method works.
3
How do the tested compounds, especially the most effective ones, Chenopodium quinoa Willd bitter seed husks and the flavonoid naringenin, modulate the binding of SARS-CoV-2 SpikeRBD and ACE2?
It will be understood better if an illustration is provided.
4
Keywords
Please be more specific. Add more keywords.
5
Abstract
Please provide more details for the tested natural compound classes:
Flavonoids, quinone, etc.
6
Please also provide the backgrounds of natural compounds in the potential of preventing the invasion of SARS-CoV-2 SpikeRBD.
Comments on the Quality of English LanguageThe authors should check thoroughly the English language, and correct any typing errors.
Reviewer 2 Report
Comments and Suggestions for Authors
Dear Authors,
The article “Microfluidic Diffusional Sizing applied to the study of natural products and extracts that modulate the SARS-CoV-2 SpikeRBD/ACE2 interaction” describes a new methodology, Microfluidic Diffusional Sizing (MDS), to measure interaction between potential antiviral or anti-inflammatory drugs with the viral target.
The issue is important and actual, the data obtained are accurately presented.
There are some comments, which can help for better understanding of the article, the aims and scopes.
1. It is written in the Abstract “recently developed Microfluidic Diffusional Sizing (MDS)” method. Who was the creator of this method is not clear : the authors of this article or the Laboratory “Fluidic Analytics, 312 Cambridge, UK”. There is no reference in the Section 4.3.
2. There is very important issue which is absent in the article – what the solvent have been used for the control samples? DMSO? If so, at what concentration and it have influenced the parameters tested.
3. The choice of the substances tested is not sufficiently justified. Some more explanation of this choice with some experimental data and references that confirm their pharmacological efficiency as antiviral or anti-inflammatory agents should be added.
4. What was the reference drug also is not clear. The reference drug must have clinically confirmed antiviral effectiveness against SARS-CoV-2. (Favipiravir , Remdesivir, Enisamium ……)
5. Dissociation constants, which show the ability of the tested substances to modulate the SARS-CoV-2 SpikeRBD/ACE2 interaction are high enough. It is needed to add discussion about these concentrations and how they can be extrapolated in vivo.
The article suits well the Journal and can be published after the correction.
Reviewer 3 Report
Comments and Suggestions for Authors
The paper by Fauquet et al., describes how natural products and extracts can mobulate the SARS CoV 2 Spike RBD/ACE2 interaction.
I find the theme very interesting but I think that the paper can be improved and the result can be easily understandable. So I suggest some changes.
Introduction is very informative only about the mechanism of action of the virus and for the MDS although I think that the information about the use of natural products (isolated compounds or/ and extracts from plants) and how they can be used in the treatment of some virus is missing.
Moreover, I am not sure which are the isolated compounds and the extracts that were tested. In fig 3 there are some compounds. Some of them are inside an cycle. I can not understand if you have detect these compounds in the extracts or don't.
If you have detect them give some information about the HPLC program that you use.
Round 2
Reviewer 1 Report
Comments and Suggestions for Authors
Dear Authors,
Thank you for your response.
The revised version of the manuscript has shown significant improvement.
The issues raised during the first review has been sufficiently addressed.
The revised version is fine.
A minor issue:
Keywords:
I would prefer the previous keywords, if, as claimed by the Authors, they should be limited to 3 ones as required by the Journal.
Thank you.
Reviewer 2 Report
Comments and Suggestions for Authors
Dear Authors, the article have been significantly improved.
I could not find the answer to the point 3 of my review, because it`s written in the Replies:
ANSWER: Please see the answer to question 6 of Reviewer 3.
But I hope, that there have been done appropriate corrections.
The article can be published.
Reviewer 3 Report
Comments and Suggestions for Authors
Dear authors,
You do a great job with your study. The current revised version is very informative and clearly presents the aim and the results of your ressearch.